# Qualitative Exploration of the ‘Rolling Unmasking Effect’ for Downwind Odor Dispersion from a Model Animal Source

**DOI:** 10.3390/ijerph182413085

**Published:** 2021-12-11

**Authors:** Donald W. Wright, Jacek A. Koziel, David B. Parker, Anna Iwasinska, Thomas G. Hartman, Paula Kolvig, Landon Wahe

**Affiliations:** 1Don Wright & Associates, LLC, Georgetown, TX 78628, USA; 2Department of Agricultural and Biosystems Engineering, Iowa State University, Ames, IA 50011, USA; 3College of Engineering, West Texas A&M University, Canyon, TX 79016, USA; dparker@wtamu.edu; 4Volatile Analysis Corporation Inc., Grant, AL 78664, USA; 22titanium@gmail.com; 5Department of Food Science, Rutgers University, New Brunswick, NJ 08901, USA; hartmant@njaes.rutgers.edu; 6Moody Gardens, Galveston, TX 77554, USA; pkolvig@moodygardens.org; 7Department of Civil, Construction, and Environmental Engineering, Iowa State University, Ames, IA 50011, USA; lwahe@iastate.edu

**Keywords:** odor, volatile organic compounds, environmental analysis, air sampling, simultaneous chemical and sensory analysis, prairie verbena, prehensile-tailed porcupine, Virginia pepperweed

## Abstract

Solving environmental odor issues can be confounded by many analytical, technological, and socioeconomic factors. Considerable know-how and technologies can fail to properly identify odorants responsible for the downwind nuisance odor and, thereby, focus on odor mitigation strategies. We propose enabling solutions to environmental odor issues utilizing troubleshooting techniques developed for the food, beverage, and consumer products industries. Our research has shown that the odorant impact-priority ranking process can be definable and relatively simple. The initial challenge is the prioritization of environmental odor character from the perspective of the impacted citizenry downwind. In this research, we utilize a natural model from the animal world to illustrate the rolling unmasking effect (RUE) and discuss it more systematically in the context of the proposed environmental odorant prioritization process. Regardless of the size and reach of an odor source, a simplification of odor character and composition typically develops with increasing dilution downwind. An extreme odor simplification-upon-dilution was demonstrated for the prehensile-tailed porcupine (P.T. porcupine); its downwind odor frontal boundary was dominated by a pair of extremely potent character-defining odorants: (1) ‘onion’/‘body odor’ and (2) ‘onion’/‘grilled’ odorants. In contrast with the outer-boundary simplicity, the near-source assessment presented considerable compositional complexity and composite odor character difference. The ultimate significance of the proposed RUE approach is the illustration of naturally occurring phenomena that explain why some environmental odors and their sources can be challenging to identify and mitigate using an analytical-only approach (focused on compound identities and concentrations). These approaches rarely move beyond comprehensive lists of volatile compounds emitted by the source. The novelty proposed herein lies in identification of those few compounds responsible for the downwind odor impacts and requiring mitigation focus.

## 1. Introduction

Without conscious effort, the majority of the human population can make the association between some characteristic environmental odors and the specific chemicals that are primarily responsible for those odors. A mother’s recognition of ammonia, without analytical confirmation, as the specific chemical that is responsible for the ‘ammonia’ odor in the vicinity of an incubating pile of urine-soaked diapers, is a simple manifestation of that innate ability. The primary factor separating the general population from sensory professionals, tasked with deconstructing complex odors, is the number and obscurity of such associations that can be made, in advance of analytical confirmation. This ability is a simple manifestation of odorant prioritization that is the basis of the research effort reported herein [1].

### 1.1. Is Solving the Downwind Odor Problem Possible without an Exhaustive List of Identified Compounds Emitted from a Source?

Concerning contemporary environmental odor issues, odorant prioritization does not appear widely recognized or referenced. It is still common to encounter references to odor issues as approximately correlated to extensive inventory listings of volatile chemicals that are shown to be emitting from ‘suspect’ odor sources. Sometimes the listing represents an extensive and complex emission mixture, potentially encompassing hundreds of volatile organic compounds (VOCs) and many chemical functionalities. These complex VOC listings are often distributed between (a) organic sulfides; (b) ketones; (c) aliphatic aldehydes; (d) aromatic aldehydes; (e) aromatic hydrocarbons; (e) terpenes; (f) alcohols; (g) volatile fatty acids; (h) hydrocarbons; (i) chlorinated hydrocarbons and (j) aliphatic siloxanes; such as in the case of odor emissions from one small sewage treatment facility. Unfortunately, these extensive lists often include a preponderance of compounds with little, if any, downwind odor impact beyond the source fence-line. More significantly, these extensive inventories often fail to include or identify the specific odorant or odorants, which are primarily responsible for the downwind odor.

### 1.2. Proposed Solution—Based upon Simplification through Downwind Odorant Prioritization

We propose solving environmental odor issues by utilizing troubleshooting techniques developed for the food, beverage, and consumer products industries. Our experience has shown that an odorant impact-priority ranking is definable for virtually every odor source, whether natural or human-made. While the composition of environmental odors, as detected by human receptors, carries the potential for extreme complexity, the reality is that there is a high degree of compositional simplification, which typically develops with increasing distance separation from the odor source. This compositional simplification can also manifest itself as changes in odor character (i.e., ‘what it smells like’). We refer to these two effects as the Rolling Unmasking Effect (i.e., RUE). The initial challenge is the prioritization of environmental odor character from the perspective of the impacted citizenry downwind. The novelty proposed herein lies in identification of those few compounds responsible for the downwind odor impacts and requiring mitigation focus.

### 1.3. Objective

In this research, we utilize a natural model from the animal world to illustrate the rolling unmasking effect (RUE) and discuss it more systematically in the context of the proposed downwind environmental odor prioritization approach. This research focused on the South American prehensile-tailed porcupine (P.T. porcupine; *Coendou prehensilis*), selected for this study due to its reputation within the zoo-keeping community as being particularly odorous and reflecting a particularly unique odor character. The ultimate significance of the proposed RUE approach is the illustration of naturally occurring phenomena that explain why some environmental odors and their sources are challenging to identify and mitigate using the analytical-only approach (focused on compound identity and concentrations only), rarely moving beyond comprehensive lists of compounds emitted by the source.

### 1.4. Rationale

The logistics involved in carrying out an odorant prioritization assessment can be challenging when targeting large area odor sources (e.g., industrial or confined animal feeding operations, CAFOs) due to the size and the downwind reach of the odor plume. However, the P.T. porcupine source model, described herein, utilizes the significant reductions in plume size and distance of reach to illustrate how the RUE approach could be applied for focusing larger-scale odor issues.

In addition, the RUE process is the same regardless of the source and scale of downwind reach. In each case, the sources’ VOC profile is typically highly complex near the source but is simplified, through dispersive dilution, as it reaches the odor frontal boundary. This simplification is reflected in both the impact-priority subset composition and the total number of odorants essential for inclusion in that subset.

Several advantages are believed reflected in selection of the P.T. porcupine as the natural odor-source model: (1) the P.T. porcupine source is publicly accessible to the readership (with varying degrees of accessibility; depending upon geographical location and access to public zoo exhibits); (2) most of the readership audience is equipped with sensory abilities that are equal to or better than those of the lead investigator and collaborators and (3) therefore can perform their own informal assessments and determine if they are in agreement with the odor-character descriptions as applied by the collaborators.

## 2. Background

### 2.1. Challenges to the Current State of the Art in Downwind Odor Assessment

Environmental odor issues can be confounded by many analytical, technological, and socioeconomic factors. While considerable know-how and technologies exist for industrial source odor mitigation, they are often not adapted for rural and agricultural odor [2]. Source-to-receptor separation can aid in lowering downwind odor impact. The dispersion has historically been described as ‘downwind dilution’ and monitored by standard techniques such as dynamic dilution olfactometry [3,4,5]. In reviews of international odor regulations [6,7] it was reported that almost all standards are based upon odor concentration limits by forced-choice olfactometry, reflecting either; (a) laboratory olfactometer based odour concentration units per cubic meter; (b) triangle bag-based odour index threshold value measurement or (c) [6] field olfactometry-based offensiveness measurement. There were no major international entities that referenced the use of chemical-analysis based methods (i.e., GC-MS, GC-Olfactometry, odorant prioritization) for environmental odor assessment, monitoring or mitigation. Still, odor sample loss problems have been identified [8,9] for some sampling devices such as Tedlar gas sampling bags [8] and, in some situations, shown incrementally improved recovery [9] through contact surface chemistry modification.

There is also broad recognition of a challenge to link specific compounds to resulting downwind odor [10,11]. In one notable example from an odorant prioritization study to the rendering industry [12], just two odorants (trimethylamine (TMA) and dimethylsulfide (DMS)) were identified as the impact-priority odorants downwind of a fish meal processing plant. This finding stands in marked contrast to an earlier study of the same issue [13], reporting ~300 organic compounds, 40 of which were odorous and stating that ‘odorous compounds included alkanes, alkenes, ketones, hydrocarbons, alcohols, alkyl halides, fatty acids, amines, aromatics, aldehydes, and epoxides’. It should be noted that this 300-compound listing did include TMA and DMS, but those were not prioritized. In a more recent study [14,15], these authors were able to identify the specific chemical odorant that is believed primarily responsible for the reported ‘skunky’ odor downwind of dense cannabis-growing operations. The team utilized an analytical approach (air sampling with solid-phase microextraction, SPME; and analysis on a gas chromatography—mass spectrometry—olfactometry system, GC-MS-O), leaf enclosure study and field observation, to isolate, identify, measure and ultimately conclude that the compound 3-methyl-2-butene-1-thiol (i.e., 321 MBT), was the primary source of this ‘skunky’ odor of cannabis [14,15]. Historically, this ‘skunky’ downwind odor has often been tied to terpenes. The 321 MBT reported discovery as the actual link with ‘skunky’ cannabis supports the more persuasive expectation of a sulfur component within the emission profile of cannabis [14,15].

The odor activity value (OAV) concept has been used with some success to show that compound-specific odor detection thresholds (DTs) can be helpful to explain why some compounds are more impactful odorants than others, with respect to CAFO housing buildings [16], cannabis storage [17] and illicit drug [18,19] odor sources. The use of simultaneous chemical and sensory analyses has also gained acceptance as a technology for isolating, ranking, and prioritizing odor-causing compounds in a complex mixture of gases. This has been illustrated for diverse odor sources such as swine barn [20], dairy manure [21], swine manure [22], and swine & dairy housing [23] sources. For example, this has been shown for *p*-cresol as a ‘signature’ downwind odor from confined animal feeding operations (CAFOs), recognizable at a great distance from the source. This prominence has been described for large swine CAFOs [24], in one case remaining the single, most offensive characteristic compound as far as 16 km away from a beef cattle feedlot [25].

Downwind odor dilution, while helpful for alleviating an odor issue, can also challenge understanding and mitigation of environmental odor impact issues. In many cases, the highest impact odorants downwind reflect concentration levels far below electronic detection limits. Further, our experience has shown that the odor ‘character’ (defined as a descriptor of what it smells like) from an environmental source can depend upon the downwind distance from that source, potentially radically different nearest the source relative to locations farther removed. This has been illustrated through synthetic odor match formulation development [26], diverse animal odor studies [27], swine CAFO downwind odor profiling [28], cattle feedlot odor source studies [24], and downwind odor from swine finisher and beef cattle operation [25].

With increasing distance separation from the source, the process of environmental odorant prioritization can be described as a rolling unmasking effect (RUE), as shown in Figure 1.

### 2.2. Natural Examples of RUE

Several examples of the RUE have been described over the past two decades. Examples are (1) odor from the large colony of Mexican free-tailed bats (i.e., *Tadarida brasiliensis*) [27,29] and (2) odor from a large cattle feedlot [28] and swine CAFO [24]. For the Mexican free-tailed bat colony [29], three distinct odor boundaries (Figure 1) were definable relative to the cave source: (i) an overpowering ‘ammonia’ odor within the cave and for ~15 m downwind of cave openings; (ii) emergence of a composite ‘rat nest’ odor, which was dominated by a member of quinazoline family, upon the decline of the masking by ammonia, and (iii) emergence of the characteristic ‘bat cave,’ ‘taco shell’ odor, dominated by 2-aminoacetophenone, upon approach to the outer ‘odor frontal boundary’; enabled by the decline of odor masking by the quinazoline odorant. Similarly, for large cattle feedlot [28] and swine CAFOs [24], at least two distinct odor boundaries were definable (i) a strong ‘fishy’/‘amine’ odor dominated by TMA within the feedlot and for several hundred meters downwind, and (ii) the emergence of a ‘barnyard’ odor, dominated by *p*-cresol, upon approach to the outer ‘odor frontal boundary’; enabled by the associated decline of downwind odor masking by TMA.

## 3. Materials and Methods

### 3.1. Odorant Prioritization Procedural Summary Outline

The general experimental process for odorant prioritization, as applied to the model environmental odor source, can be summarized as the Steps 1–7 described below. In this research and manuscript, only Steps 1–5 (qualitative) are applicable (Figure 2).

The complete series of 1–7 steps are presented as a general overview of qualitative and quantitative process:**Step 1**—**Downwind composite odor assessment**—qualitative, at-site odor-character assessment by the panelist; in this case, (D.W.W.), a gas chromatography-olfactometry (GC-O) investigator with 20+ years of experience, odor troubleshooting for the industry. The goal of this stage is to observe recognizable odors that are consistent and perceived as characteristic of the source, at the downwind outer boundary and at the time of the at-site assessment.**Step 2**—**GC-O-based odorant prioritization**—qualitative, on-instrument assessment by the panelist (D.W.W.); attempting to make a connection between the observed downwind odor character and individual compounds that are perceived as character-defining for that odor.**Step 3**—**First-pass odor-match validation of impact-priority hypothesis from Step 2**—qualitative odor-match based confirmation by conference with associate GC-O investigator(s), where possible; in the case of the P.T. porcupine, an experienced GC-O investigator with 10+ years of experience (A.I.) odor troubleshooting for industry; generally involving on-instrument GC-O based crosscheck.**Step 4**—**Development of a synthetic formulation****for final odor-match-based validation**—the panelist (D.W.W.) attempts to develop a formulation, in low odor, food-grade propylene glycol carrier, which reflects a high-fidelity odor-match to that of the targeted environment downwind. This formulation can range from very simple single odorants to multi-odorant blends, matching the odorant concentration ratios existing in the targeted environments downwind.**Step 5**—**Final odor-match validation of impact-priority hypothesis from Steps 1 and 2**—qualitative or quantitative odor-match-based validation by conference with volunteer sensory panelists drawn from (**a**) downwind citizenry; (**b**) other community stakeholders or (**c**) professional sensory panel.(as required) **Step 6**—**Analytical method development targeting impact-priority odorants defined and validated; Steps 1 through 5**—quantitative method development for follow-on odor investigation, monitoring, and mitigation strategy focusing.(as required) **Step 7**—**Instrument-based environmental odor monitoring based upon impact-priority odorants**—quantitative monitoring for correlating downwind environmental impact and upwind source prioritization.

### 3.2. PT Porcupine Urine Sampling

A urine sample with the entrained fecal matter was collected (see details in Appendix A). The VOCs were collected from the equilibrated headspace formed within a 1-quart glass headspace vessel containing a few drops of the urine sample, injected onto a crumpled low-odor paper towel substrate. The sample was equilibrated, stored, and sampled in an open-air laboratory environment at 24 °C. In addition, direct comparison samples were collected utilizing a single, designated, 1 cm × 75 µm^−1^ Carboxen/PDMS SPME fiber (Supelco, Bellefonte, PA, USA). SPME fiber insertion into the headspace was through a pinhole placed in the vessel’s PTFE closure. The amount of extracted VOCs was varied by altering the time the SPME fiber was exposed to the equilibrated headspace.

### 3.3. PT Porcupine Exhibit—Downwind Air Sample Collections with SPME

A series of direct environmental air samples were collected with SPME. The SPME fibers were: (1) preconditioned at 260 °C; (2) transported, under dry-ice storage conditions, to the Moody Gardens Rainforest for the VOC collection by direct SPME fiber exposure within the P.T. porcupine indoor exhibit and (3) return of samples to the laboratory under dry-ice storage conditions. Preconditioned SPME samplers were secured onto a field-support fixture within the exhibit enclosure; the adsorbent coated fiber tips extended from their protective needle sheaths (i.e., exposed to the enclosure environment). SPME exposures to air were executed for 7 and 9 min, respectively. Duplicate SPME fibers were exposed for 15 h. Finally, the four sample collections were transported, under dry-ice conditions, back to the laboratory for odorant prioritization assessment. At the time of sample collection, the smell, far downwind from the enclosure, was described as distinct ‘grilled onion’.

### 3.4. Multidimensional Gas Chromatography-Mass Spectrometry-Olfactometry

Simultaneous chemical and sensory analyses combined olfactometry (O), multidimensional (MD) separation techniques with conventional GC-MS instrumentation. An MDGC-MS-O system was used for odorant prioritization. The system consisted of an Agilent 6890 GC/5975B MS modified for MDGC-MS-O utilizing an AromaTrax™ control system (Volatile Analysis Corp., Round Rock, TX, USA). Details regarding general hardware and operation have been described elsewhere [22,30]. Specific operational parameters were as follows: injection mode: split-less with solid-phase microextraction (SPME) sample collection and delivery; the SPME fibers were left in place once they were inserted, achieving: (i) the initial adsorbed VOC injection for the current analysis and (ii) preconditioning the fiber for the subsequent sample collection event; injection temperature: 250 °C; detector #1: FID (280 °C); detector #2: Agilent 5975B MSD in MS-SCAN or -SIM acquisition modes; column #1: 12 m × 0.53 mm ID BPX 5-1.0 µm film (pre-column from SGE); column #2: 25 m × 0.53 mm ID BPX 20-1.0 µm film (analytical column from SGE); column temperature program (overview survey and MDGC-MS-O): 40 °C initial, 3 min hold, 7 °C∙min^−1^, 220 °C final, 20 min hold.

### 3.5. MDGC Parameters for Compound Isolation with Heart-Cutting

Concerning MDGC heart-cut isolation/clean-up of the two target ‘onion’ odorants for the P.T. porcupine; (1) optimal band for heart-cut #1 (i.e., unknown ‘onion’ odorant #1) was ~9.9 to 11.2 min; (2) optimal band for cryotrap #1 was ~9.4 to 11.5 min; (3) optimal band for heart-cut #2 (i.e., unknown ‘onion’ odorant #2) was ~14.4 to 15.8 min; (4) optimal band for cryotrap #2 was ~13.9 to 16.1 min; (5) long SPME collection of the whole urine headspace yielded overwhelming odor responses but no obvious associated mass spectral ion detail for the critical ‘onion’ odorants.

### 3.6. Chemical Identification

The MS was operated in MS-SCAN mode for survey mode odorant identification. The mass range (35 to 400 amu) was scanned at 3.84 scan∙s^−1^. The resulting spectra were analyzed with Benchtop PBM software, referencing the Wiley 7 library for the best-match ranking against the database. The panelist retained final over-ride determination as to the likelihood of correctness of the best-match listings. Spectra without a suitable library match were considered ‘unknown’; unless overridden by considerations of known retention time combined with simultaneous odor character recognition at the sniff port.

Unfortunately, the panelist (D.W.W.) was unable to confirm the chemical identities of the two character-defining ‘grilled onion’ odorants from the P.T. porcupine environments. Therefore, in a further attempt to identify these unknowns, collaborations with experts in the food flavor/aroma field were engaged (#1 T.G.H.; #2 A.I). Their approaches are summarized in detail in the Appendix A. Briefly, collaborator #1 used purge-and-trap thermal desorption followed by GC-MS-based analyses. Collaborator #2 used the same MDGC-MS-O approach as the panelist and served as a crosscheck of the proposed VOC/odorant identity profiles.

## 4. Results

The model case study below serves as an illustration of the RUE concept (Table 1). This case documents an apparent (i) difference between the overall odor perceived at the source and the downwind odor frontal boundary, accompanied by (ii) chemical and odor simplification. The ultimate goal is to use RUE to correlate the downwind odor with the individual chemical(s) most responsible for that odor.

### 4.1. Case Study: Prehensile-Tailed Porcupine

#### 4.1.1. Initial Odor Assessment at the Source and Downwind

The panelist first encountered the smell at the downwind odor frontal boundary (Figure 3) from the P.T. porcupine enclosure, detecting a very distinct and familiar ‘grilled onion’ and ‘1950s hamburger joint’ character. The panelist initially walked upwind to determine where the food court must be located. However, upon walking deeper into the odor plume, the panelist encountered, almost simultaneously, an intense ‘foul’ odor and an associated exhibit display sign which read, ‘*What is that Foul Odor?*’ The sign heading was followed by a description of the P.T. porcupine exhibit as the source (Appendix A).

The near-source smell was perceived as ‘phenolic,’ ‘industrial,’ and ‘foul.’ The dramatic difference in character was particularly surprising considering that only a few paces separated the pleasant ‘grilled onion’ at the odor frontal boundary and the ‘foul’ odor deeper into the plume.

The working hypotheses are:(1)*due to the remarkable similarity in downwind odor characteristics, there is assumed to be some chemical compound commonality between the priority odorant subsets for the P.T. porcupine and typical ‘grilled onion’ odors*. Likewise,(2)*due to the remarkable dis-similarity in downwind odor characteristics, there is assumed to be some chemical compound disconnect between the priority odorant subsets for the P.T. porcupine and typical swine CAFO odors*.

The driving questions were, therefore:Are there common character-impact odorants to both the P.T. porcupine and typical ‘grilled onion’ sources that account for the striking similarity in composite odor at their respective odor frontal boundaries?Are there character-impact odorants for P.T. porcupine and swine CAFO sources that account for the striking difference in composite odor character at their respective odor frontal boundaries?What is the overall agreement between the P.T. porcupine and swine CAFOs when comparing their minimum priority odorant subsets downwind with their full (at source) odorant and underlying VOC profiles?

#### 4.1.2. Odorant Prioritization

The MDGC-MS-O-based odorant prioritization confirmed the pre-analysis assumption that the impact-priority odorant would be found to be traceable to a specific homolog from the extensive ‘onion’ odor allylic-polysulfide family [31,32]. The ‘1950s hamburger joint’ odor note had previously been isolated and described (i.e., by retention time and ‘sniff port’ detector basis only) in prior onion-sourced odorant prioritization investigations. Upon inspection of P.T. porcupine urine headspace, the ‘grilled onion’ odor note eluted a few seconds before dipropyl trisulfide and earlier still than the propyl—propenyl trisulfide isomer series @20.8 min. A second ‘onion’ note (i.e., unknown ‘body odor’, ‘onion’ @13.9 min) was also found. Remarkably, the extremely complex headspace odor profile appeared free of other members from the onion-sourced allylic-polysulfide family (Figure 4 and Figure 5).

These two character-defining ‘onion’ odorants were shown to emerge from a vast and complex odorous VOC field, common to mammalian waste, e.g., North American porcupine [33], scent-marking of wild cats [34], marking fluids of Siberian tiger [35], and lion [36]. The chemical identification effort for the two character-defining, ‘onion’ odor compounds included three alternative approaches. Despite these considerable efforts, the identifications of the two character-defining, ‘onion’ odors remained elusive. The likely reason is these two odorants’ extreme trace concentration and odor potencies (Figure 6, Figure 7 and Figure 8). In addition, work to date suggests that the targeted unknown ‘onion’ carrier compounds are not related to specific polysulfide odorants, previously reported as responsible for ‘grilled onion’ and ‘fried onion’ odor character [1,37,38].

#### 4.1.3. Contrasting Downwind Odorant Prioritizations—The P.T. Porcupine vs. a Swine-Barn CAFO

The VOC emission composition for P.T. porcupine and swine barn have much in common (Table 2). The P.T. porcupine, like the CAFOs, presents with significant emission loadings of the reduced sulfurs, free-fatty acids, indolics, and phenolics (i.e., including *p*-cresol), all of which factor heavily in CAFO emission profiles. The absence of CAFO-like odor character from these odorous VOCs at the P.T. porcupine’s odor frontal boundary magnifies the impact significance of the two unknown ‘onion’ odorants as the characteristic ’grilled onion’ odor carrier. It is particularly interesting that the P.T. porcupine and swine barn sources generate distinctly different odor characteristics at their respective odor frontal boundaries, despite sharing much in common through their VOC emission profiles at the source.

The P.T. porcupine-based odorant prioritization process was carried through to Step 5 validation as outlined above. The illustration of the responses by panelists is summarized in Table 3. However, proceeding on to Step 6 (quantitative instrument-based monitoring protocol development) would have to await pre-concentration (e.g., thermal desorption) based sampling and odorant identification/detection; required steps if this investigation had been tied to an actual environmental odor issue. These follow-up efforts would focus on the two, as yet unidentified and ‘character-defining’, ‘grilled onion’ odorants for odor monitoring and mitigation assessment purposes.

While the natural-model odor source selected for focus in this manuscript is the prehensile-tailed porcupine, the authors have also carried out detailed odorant prioritization assessments of two additional natural odor sources from the plant world; prairie verbena and Virginia pepperweed (both indigenous to Central Texas). While not carried through to the same level of validation as the P.T. porcupine, they do present additional insight into the odorant prioritization process. These results are summarized in the Appendix A.

## 5. Discussion

### 5.1. Implications of the Rolling Unmasking Effect and Odorant Prioritization for Environmental Odor Mitigation and Monitoring Strategy Development

These results illustrate an important consideration and a model for larger-scale community environmental odor sources. To date, the source is often the focus when challenged with solving an environmental odor issue. While it is essential to do so when addressing downwind odor impact, it is possible to ‘look’ too closely at the source. Focusing on all compounds present at the source often expands the study to include background noise, an unnecessary expenditure if the goal is to reduce downwind environmental odor impact.

It is recommended to initially focus on the smallest subset of odorous chemicals, representing significant impact and downwind reach. This simplification was illustrated by the two unknown ‘onion’ odorants responsible for the ‘grilled onion’ odor downwind of the P.T. porcupine source, which emerged from a complex VOC background emission at the source. The character-defining odorants are first recognizable at the odor frontal boundary. The remaining complex odorous ‘noise’ near the source is often eliminated through the natural dilution process in migrating downwind. Success in mitigating the highest impact odorants results in moving the odor frontal boundary back toward the source, reducing its outward reach (i.e., the most efficient approach to developing effective remediation, monitoring, and mitigation strategies).

The reduction in downwind reach could be realized by selective elimination of only the few character-defining compounds. However, as was shown in Figure 1, the reduction in downwind odor reach will actually be determined by the distance separation between the outermost frontal boundary and the nearest secondary boundary in retreating upwind toward the source. With regard to such a ‘hypothetical’ selective elimination strategy, the best-case scenario would be a considerable distance separation between the frontal boundary and the closest secondary boundary. This may or may not be what actually exists relative to a source. The natural, steady-state profile for a source could reflect a relatively narrow separation between frontal and secondary boundaries. It is possible to encounter an unintended consequence of the selective elimination of only the (one) character-defining odorant that is responsible for the odor at the frontal boundary. The result could be the ‘unmasking’ of another odorant with odor character even more offensive than the original. An example of this is potentially reflected in the P.T. porcupine odor. The selective elimination of only the ‘grilled onion’ character-defining odorants would elevate the rest of the impact-priority odorant subset. The emerging secondary odor boundary could be more ‘barnyard’ in character [26,27], owing to the *p*-cresol prominence and the overall odor profile similarity between P.T. porcupine and swine barn (summarized in Table 2). Therefore, a more realistic strategy is to focus mitigation on the smallest character-impact subset of odorants responsible for frontal boundary and near-source odor character. For example, in the case of the P.T. porcupine, that smallest combined impact-priority subset consists of the 5 to 7 odorants leading Table 2.

### 5.2. Counter-Intuitive Odor Masking

The impact-priority rankings can be counter-intuitive, revealing an unexpected difference in odor-defining compounds near-source vs. frontal boundary. The OAV concept has been historically applied to gauge the difference in odor potency between different odorous compounds emitted from a source. OAV is defined as the ratio between an odorant’s concentration and the odor threshold concentration of that compound. Unfortunately, the concept fails to adequately explain the apparent difference in odor dominance by compounds when comparing near-source versus at-distance odor receptor sites. This is because it assumes relatively constant OAV values spanning the time and source-to-receptor distance (which, in practice, does not always hold).

An alternate representation of this observed counter-intuitive unmasking effect is proposed (Figure 9) based on the earlier observations of bat colonies [27,29]. This is a graphical representation of the odor sigmoid intensity curves for two competing odorants, reflecting relatively high vs. relatively low odor potencies; 2-aminoacetophenone and ammonia, respectively. The concentration vs. odor intensity is delineated by (1) odor threshold value—the minimal concentration that a human receptor can detect as a perceptible odor change; (2) odor recognition threshold value—the minimal concentration that can be detected and recognized by the human receptor as to odor character/odor source and (3) odor saturation threshold value—the concentration level at which all related olfactory receptors are activated and above which any additional concentration increase will fail to induce a corresponding increase in response intensity. While ammonia requires a much higher concentration to exceed the odor and recognition thresholds, once exceeded, it rises to a response level that overtakes 2-amino acetophenone. Thus, while OAV values account for the dominance of a higher-impact odorant up to being masked, it fails to account for the apparent reversal in dominance above that juncture (Figure 8). Several mechanisms have been proposed for this observed non-linearity of the OAV values at higher concentrations, including (1) synergistic effects, (2) receptor blocking effects, and possibly others.

Another example of counter-intuitive odor masking is ‘musty’ cork taint in the wine industry caused by 2,4,6-trichloroanisole (TCA) and tribromoanisole (TBA). The TCA’s ‘*faint odor similar to acetophenone*,’ as described in the 13th Edition Merck Index [39], is noteworthy, considering its published odor threshold value of 10 parts per trillion [40]. By comparison, that for TBA is ~30 ppq [41]. The odor response to TCA or TBA contamination can be initially masked by many other common odors co-emitting from a source, regardless of relative odor potency. In contrast, however, TCA and TBA will almost always remain the ‘cork taint’, ‘musty’ defining compounds after all others have weathered away to levels below their respective odor ‘masking’ effect.

### 5.3. Implications of the RUE for Community Environmental Odor Issues

Understanding the RUE process can facilitate improved communication between critical stakeholders to a community environmental odor issue. Historically, the downwind citizenry has been least effectively represented in community discussions regarding odor assessment, chemical prioritization, odor monitoring, and mitigation strategy development. The communication challenges can be illustrated, at least partly, by drawing parallels from the sense of visual color perception. For example, the pictured cube (Figure 10), when presented to a human sensory panel and asked to describe the color, should elicit an overwhelming response as ‘red’. If then asked to expand on this assessment, the panelists might add various descriptor modifiers, such as red as ‘tomato’, ‘blood’, and ‘fire engine’. These modifiers would likely reflect a considerably lower level of consensus since they are cultural and/or personal experience based. Fortunately, material color-wheels can effectively neutralize these biases and reconcile the modifying descriptors to a consensus.

In contrast, for the sense of smell, we are limited solely to such subjective descriptor modifiers for reconciling communication between stakeholders regarding odors of common interest (e.g., ‘sewer-like’, ‘barnyard-like’, ‘skunky’, ‘musty’). Sensory professionals representing selected industries have developed odor/aroma/flavor wheels that attempt to emulate the color wheel for drinking water [42] and beer [43]. While these sensory wheels can be very effective tools in reconciling discussions between trained sensory professionals, they are too cumbersome for practical use by lay panelists (e.g., downwind citizenry). The practical challenge for such odor wheels is that they, too, rely on relatively vague descriptors such as ‘musty’, ‘barnyard’, ‘earthy’.

The simplification of odor profiles, induced by the RUE, opens up the possibility of introducing a reconciling tool for the odor that is more closely aligned with the simplicity of color and the color wheel. This tool uses chemical odor-matching [26,27]. For example, reconciling the communication regarding odor-character at their respective odor frontal boundaries is simplified by having a sensory panelist either confirm or reject a proposed odor-match using ‘suspect’ high-purity reference chemicals.

The odor-match query of a lay panelist relative to a targeted environmental odor can be a simple YES or NO when presented with a trace amount of a ‘suspect’ character-defining odorant. This simplicity negates the requirement for extensive panelist training, experience, or memory acuity relative to odor recognition. Such straightforward odor-match surveys can be easily expanded to include query variations such as (1) picking the best odor-match from a multi-unknown odorant line-up, including the ‘suspect’ character-defining odorant, and (2) applying an odor-match fidelity grading estimation to a best odor-match selection. The odor-match validation process is the same whether the chemical reference is a single, character-defining odorant (e.g., dominant at the odor frontal boundary) or a multi-odorant formulation (e.g., synthetically replicating the combined frontal boundary + near-source odor character).

The odor-match approach can have some practical challenges. Even if the impact-priority odorants are isolated utilizing MDGC-MS-O, there is no guarantee that (1) a library-match based mass spectral identification of the impact-priority subset can be achieved; (2) the suspect odorous compound(s) are commercially available for synthetic odor-match blending or (3) the chemicals are available in sufficiently high purity (i.e., odor-purity). It is noteworthy that the P.T. porcupine yielded an excellent illustration of the potential challenges. Despite extraordinary efforts utilizing: (1) MDGC-MS-O based target odorant purification/separation and (2) an ‘onion’ polysulfide targeted pre-concentration enrichment protocol, the identities of the two character-defining ‘onion’ carrier odorants remain elusive. As a result, the panelist (D.W.W.) applied the following novel concept [44,45] for communicating regarding these high-impact odorant ‘unknowns**’** and ‘unavailables’.

### 5.4. DoubleHeart-Cut Isolation of High-Impact Odorants from Crude Source Materials

The novel concept [44,45] utilizes MDGC, in sample-prep mode, for purification/isolation/capture of the ‘suspect’, high-purity reference odorants from readily available crude source materials. Once refined and captured, the proposed priority odorants can be utilized offline for presentation to the lay panelists for odor-match communication or validation of impact-priority/character-defining status. For P.T. porcupine, the ‘dirty’ urine was utilized as the crude source material, and the unknown ‘onion’ odorant #1 was targeted for initial odor-match validation. An inert, low-odor, polyolefin gas-tight syringe was used to ‘vacuum’ aspirate this fraction (Figure 11), capturing the targeted unknown ‘onion’ odorant #1 peak as it eluted to the olfactory detector nose-cone. The experimentally determined heart-cut effectively isolated the targeted unknown ‘onion’ odorant #1 from the bulk of potential VOC interference peaks and odorants.

Offline composite assessment of the syringe vapor contents confirmed the odor purity of the isolated fraction and yielded consensus amongst three collaborators for the ‘onion’/‘grilled onion’ odor character descriptor. A similar agreement for the high-fidelity match to the characteristic downwind odor of the P.T. porcupine exhibit was made with members from the Moody Gardens Rainforest Exhibit team. However, it is also interesting to note that one team member did not characterize the odor as ‘onion’ specifically; instead, it had reminded her of a favorite sauce that her grandmother frequently made. The second team member called the odor character ‘stale onion’. These contrasting odor character descriptors reflect the need for reconciling the distinctly different contrasting descriptors, from multiple odor panelists, for the same chemical odorant.

## 6. Conclusions

The P.T. porcupine, processed as a small, natural odor-source model, illustrates the odorant prioritization process as a potential strategy for focusing community environmental odor issues. Extreme, RUE-driven odor simplification-upon-dilution was described for the P.T. porcupine. Its downwind odor frontal boundary was shown to be dominated by two potent, character-defining odorants (1) ‘onion’/‘body odor’ odorant #1 and (2) ‘onion’/‘grilled’ odorant #2. In contrast with its boundary simplicity, however, the P.T. porcupine source presented with considerable compositional complexity and composite odor character difference when comparing near-source versus odor frontal boundary.

Significant parallels for community odor issues can be drawn from odorant prioritization and the RUE-driven simplification-upon-dilution process, as demonstrated for the P.T. porcupine:the potential for focusing on odor monitoring strategy development to that most technologically appropriate for the impact-priority subset of odorants. Understanding that one of the two character-defining odorants was a semi-volatile (i.e., with limited volatility) we know that long-term storage of whole-air samples in plastic bags is not an option.the focusing of odor mitigation strategy development to the impact-priority subset of odorants. Understanding that one of the two character-defining odorants was a semi-volatile (i.e., with limited volatility), we know that activated carbon adsorption-based mitigation strategy becomes a more economically viable option.making possible the integration of odor-matching as a reconciling tool for improving communication, among stakeholders, regarding community odor issues.

Since the P.T. porcupine was selected as a ‘neutral’ demonstration of the odorant prioritization process, it does not reflect an actual community environmental odor issue. However, if it had been, it is noteworthy that the downwind citizenry could have been (1) made aware that the proposed impact-priority ‘onion’ odorants were present at levels that are below the detection limits of one of our most-sensitive electronic detectors; (2) these impact-priority odorants are also common to onion emission and therefore unlikely to have a high toxicological impact and (3) given the opportunity to confirm for themselves, through odor-matching demonstration utilizing their own sensory capabilities, that the proposed impact-priority hypothesis are correct. This possibility has the potential for alleviating some psychologically induced health effects, should they exist.

## Figures and Tables

**Figure 1 ijerph-18-13085-f001:**
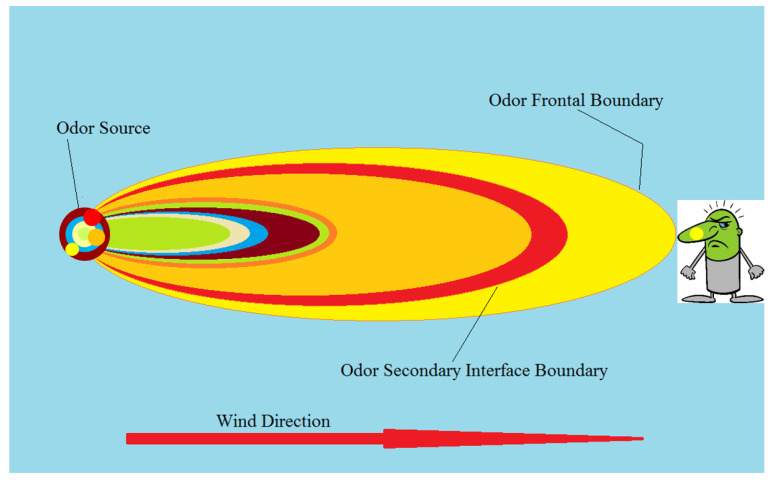
Pictorial representation of the ‘rolling unmasking effect’ (RUE). The source is a complex mixture of odorants (left), yet it is simplified to a single impactful odorant (illustrated with the yellow dot) at the receptor (right) downwind. The odor frontal boundary represents the farthest downwind reach (impact; marked with a yellow oval) of a single compound (marked with a yellow circle at the source and receptors’ nose), while the internal colored ovals represent the boundaries of sequential odor unmasking as the secondary-impact odorants are diluted below their detection/masking concentration levels.

**Figure 2 ijerph-18-13085-f002:**
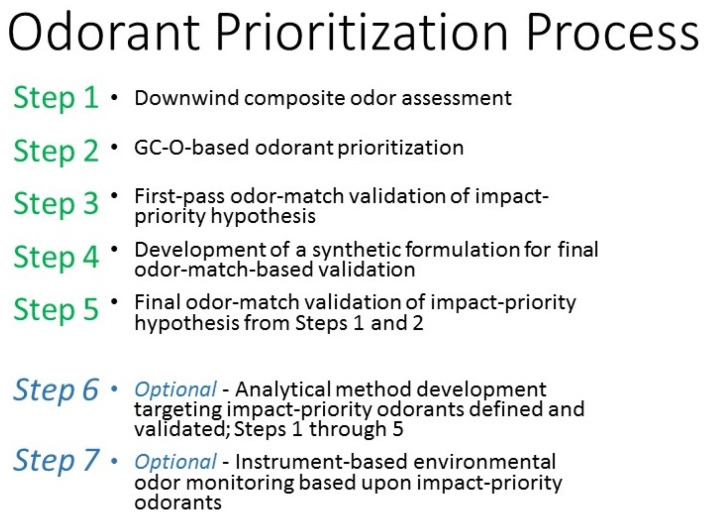
Odor prioritization process concept.

**Figure 3 ijerph-18-13085-f003:**
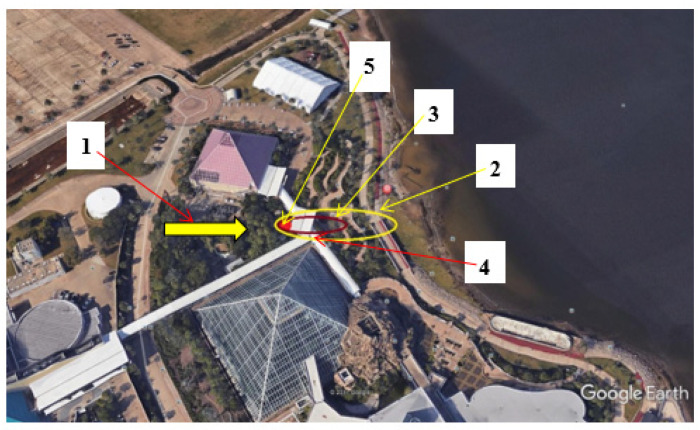
P.T. porcupine encounter in Moody Gardens. (1) Wind direction; (2) odor frontal boundary; (3) approximate secondary (near-source) boundary; (4) investigator’s approximate location upon initial encounter and (5) location of outdoor enclosure of the odor source; the P.T. porcupine. Google Earth image.

**Figure 4 ijerph-18-13085-f004:**
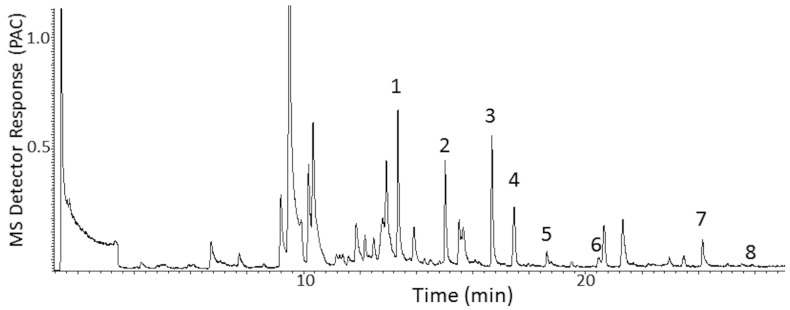
Chromatogram of the P.T. porcupine indoor exhibit chamber; total ion overview VOC profile generated in ms-SCAN acquisition mode. Volatiles collection by 15 h SPME fiber exposure to the chamber environment. 1 = acetic acid; 2 = propanoic acid, 3 = butyric acid, 4 = isovaleric acid, 5 = valeric acid, 6 = hexanoic acid, 7 = *p*-cresol, 8 = 4-ethyl-phenol. MS detector response is reported in millions of peak area counts (PACs).

**Figure 5 ijerph-18-13085-f005:**
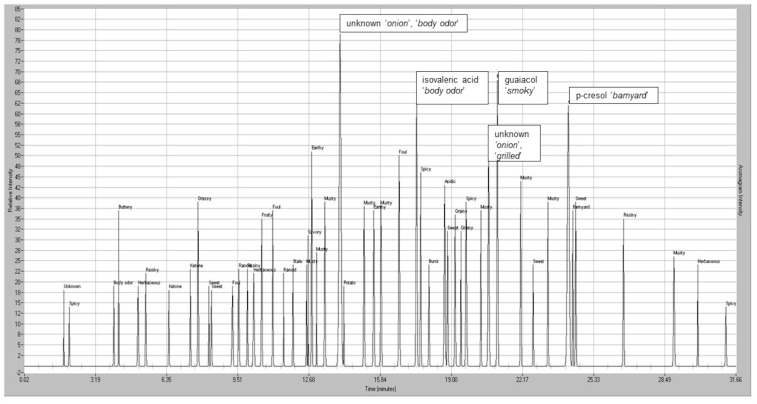
Aromagram odor profile of the P.T. porcupine indoor exhibit chamber odorants generated by GC-Olfactometry. Volatiles collection by 15 h SPME fiber exposure to the chamber environment. The five impact priority odorants stand out from the multi-odorant background noise field.

**Figure 6 ijerph-18-13085-f006:**
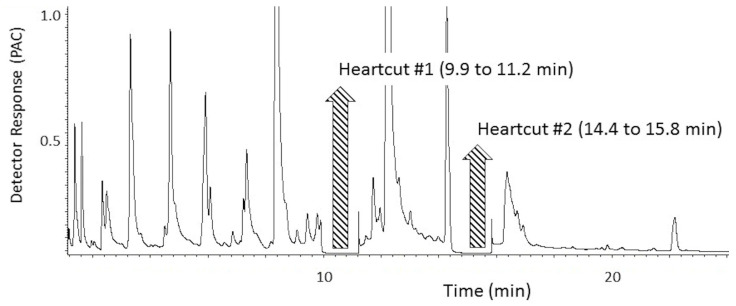
Chromatogram of male P.T. porcupine urine headspace VOCs from the pre-column separation and the FID. Two heart-cut bands targeted the ‘onion’ isolation. Volatiles collection by 10 min SPME fiber exposure. FID detector response is reported in millions of peak area counts (PACs).

**Figure 7 ijerph-18-13085-f007:**
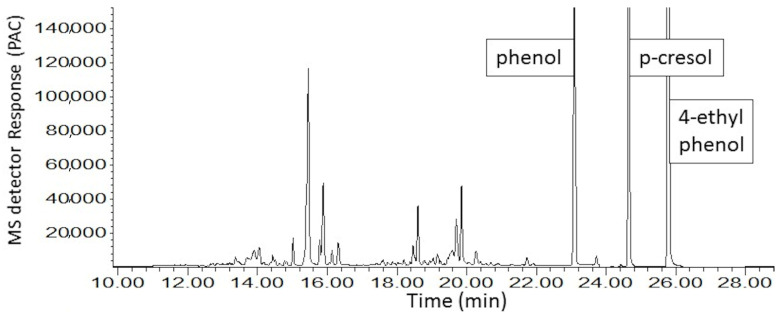
MS-TIC chromatogram of male P.T. porcupine urine headspace VOCs; analytical column separation of two ‘onion’ carrier target heart-cut isolation bands (cryotrapped). Volatiles collection by 60 min SPME fiber exposure.

**Figure 8 ijerph-18-13085-f008:**
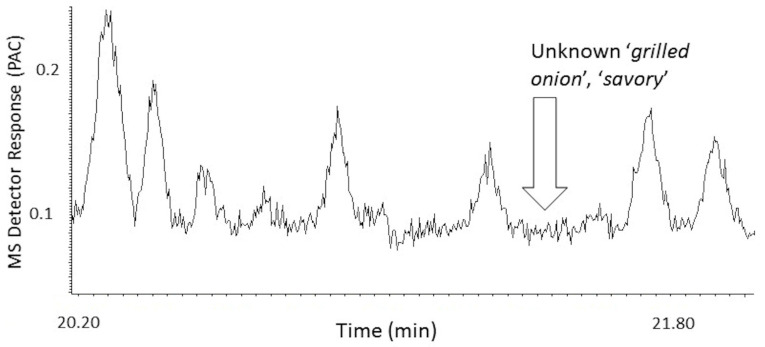
Total ion chromatogram of male P.T. porcupine urine headspace VOCs; analytical column separation focused on the second of two cryotrapped ‘onion’ carrier target heart-cut isolation bands. Volatiles collection by 10 min SPME fiber exposure. MS detector response is reported in millions of peak area counts (PACs).

**Figure 9 ijerph-18-13085-f009:**
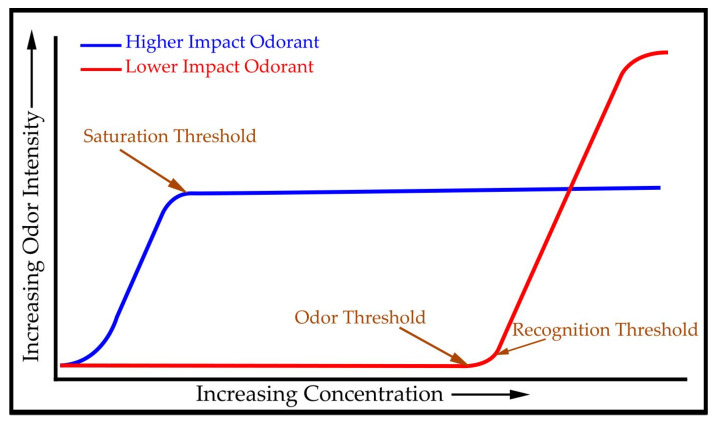
Comparison of odor threshold curves for a higher impact odorant (2-amino-acetophenone, blue line, greater reach) versus lower impact odorant (ammonia, red line, shorter reach but masking near-source) can explain why the OAV concept can fail to explain the difference between priority, odor-defining odorants downwind versus near the source.

**Figure 10 ijerph-18-13085-f010:**
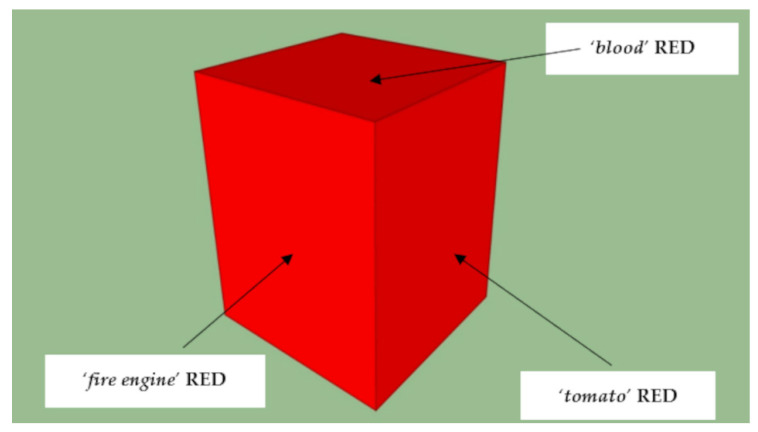
Communication about subjective odor descriptors can be more challenging than color perception.

**Figure 11 ijerph-18-13085-f011:**
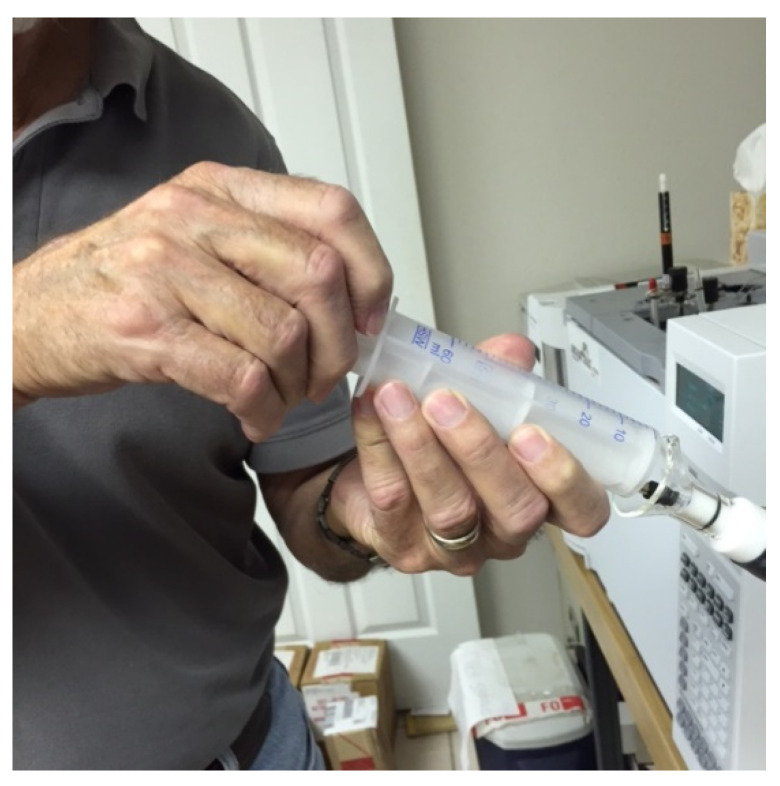
Whole air fraction collection process of high-impact odorants from the sniff port for offline odor assessment.

**Table 1 ijerph-18-13085-t001:** Guide to the matrix of experiments following the odorant prioritization procedure.

Steps	Case Study: PT Porcupine
(1) Downwind composite odor assessment	X (Appendix A)
(2) GC-O based odorant prioritization	X
(3) First-pass odor-match validation—GC-O	X
(4) Development of a synthetic odor-match formulation	X
(5) Final odor-match validation	X
(6) Analytical method development	(optional, not required)
(7) Instrument based environmental odor monitoring	(optional, not required)

**Table 2 ijerph-18-13085-t002:** Comparative impact-priority odorants; P.T. Porcupine vs. Swine Barn.

Prehensile Porcupine VOCs & Odorants *	Common Priority Odorants *	Swine Barn VOCs & Odorants *
*odor character = ‘grilled onion’*		*odor character = ‘barnyard’*
**unknown ‘onion’ @13.9 min**		
**unknown ‘onion’ @20.6 min**		
***p*-cresol**	***p*-cresol**	***p*-cresol**
**butyric acid**	**butyric acid**	**butyric acid**
**isovaleric acid**	**isovaleric acid**	**isovaleric acid**
		2-amino acetophenone
**4-ethyl phenol**	**4-ethyl phenol**	**4-ethyl phenol**
		-quinazoline
*skatole*	**skatole**	**skatole**
*indole*	**indole**	**indole**
*Sulfides*		*Sulfides*
*dimethyl trisulfide*	** *dimethyl trisulfide* **	** *dimethyl trisulfide* **
*methyl mercaptan*	** *methyl mercaptan* **	** *methyl mercaptan* **
*dimethyl sulfide*	** *dimethyl sulfide* **	** *dimethyl sulfide* **
*propyl mercaptan*		propyl mercaptan
*dimethyl disulfide*	** *dimethyl disulfide* **	** *dimethyl disulfide* **
*hydrogen sulfide*	** *hydrogen sulfide* **	** *hydrogen sulfide* **
*Fatty Acids*		*Fatty Acids*
** *valeric acid* **	** *valeric acid* **	** *valeric acid* **
** *hexanoic acid* **	** *hexanoic acid* **	** *hexanoic acid* **
** *propanoic acid* **	** *propanoic acid* **	** *propanoic acid* **
** *acetic acid* **	** *acetic acid* **	** *acetic acid* **
** *heptanoic acid* **	** *heptanoic acid* **	** *heptanoic acid* **
*Amines*		*Amines*
trimethylamine		trimethylamine
diethylamine		diethylamine
1-pyrroline		1-pyrroline
*Aromatics*		*Aromatics*
** *guaiacol* **	** *guaiacol* **	** *guaiacol* **
** *benzaldehyde* **	** *benzaldehyde* **	** *benzaldehyde* **
** *4-ethyl phenol* **	** *4-ethyl phenol* **	** *4-ethyl phenol* **
** *phenol* **	** *phenol* **	** *phenol* **
4-methyl-2-nitrophenol		4-methyl-2-nitrophenol
para-vinyl phenol		para-vinyl phenol
benzoic acid		benzoic acid
phenyl acetic acid		phenyl acetic acid
** *benzyl alcohol* **	** *benzyl alcohol* **	** *benzyl alcohol* **
*Ketones*		*Ketones*
2-octanone		2-octanone
		6-methyl-5-heptene-2-one
		2-undecanone
		pentadecanone
** *diacetyl* **	** *diacetyl* **	** *diacetyl* **
** *acetone* **	** *acetone* **	** *acetone* **
*Aldehydes*		*Aldehydes*
** *hexanal* **	** *hexanal* **	** *hexanal* **
** *nonanal* **	** *nonanal* **	** *nonanal* **
** *methional* **	** *methional* **	** *methional* **
		undecanal
*Alcohols*		*Alcohols*
		1-octene-3-ol
		3-octanol
		1-heptene-3-ol
		trans-farnesol
		maltol
		geosmin
*Miscellaneous*		*Miscellaneous*
** *2-methyl furan* **	** *2-methyl furan* **	** *2-methyl furan* **
1,3-pentadiene		2-pentyl furan
dimethyl pyrazine		dimethyl pyrazine
4,8-dimethyl-1,3,7-nonatriene		acetamide
1-methoxy-1,3,5-cycloheptatriene		4-methyl pyridine
tridecane		propanamide
		6-heptyltetrahydro-2H-pyran-2-one
		butanamide
		3-methyl-phenyl acetate
		phenyl ethyl alcohol
		pentamide
		2-pyrrolidinone
		hexadecane
		valerolactam
		5-methyl-2,4-imidazolidinedione

Notes: * many chemical identifications, beyond the impact-priority compounds, should be considered as tentative; they are the product of best-match efforts from Wiley and NIST mass spectral libraries matching. Many listed character-defining and character-impact odorants have been confirmed through on-instrument retention time and odor character matching. Priority odorants **bolded,**
*italics* = *minor odorants, **common odorants*** are italicized and bolded.

**Table 3 ijerph-18-13085-t003:** Summary of the 1–5 steps for odor-match validation.

Panelist	Designation	Odor Descriptor	Odor-Match?	Match-Source	Odor-Match Components
#1	lead investigator (D.W.W.)	‘grilled onion’/‘hamburger joint’	yes	environment + urine headspace + GC-O odorant isolate	environment + GC-O + GC-O odorant isolate
#2	investigator #2 (A.I.)	‘grilled onion’	yes	urine headspace + GC-O odorant isolate	urine headspace + GC-O + GC-O odorant isolate
#3	associate #1	‘grilled onion’	yes	GC-O odorant isolate	odor character description
#4	associate #2	‘grilled onion’	yes	GC-O odorant isolate	odor character description
#5	associate #3	-	no	GC-O odorant isolate	odor character description
#6	zookeeper	‘savory sauce’	yes	environment + GC-O odorant isolate	environment + GC-O odorant isolate
#7	curator (P.K.)	‘stale onion’	-	environment	not available for survey

## Data Availability

The original contributions presented in the study are included in the article; further inquiries can be directed to the corresponding authors.

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
