# Peer review of "Qualitative Exploration of the ‘Rolling Unmasking Effect’ for Downwind Odor Dispersion from a Model Animal Source"

_ijerph, 2021, doi:10.3390/ijerph182413085_

Round 1

Reviewer 1 Report

Thank you for submitting your manuscript to the International Journal of Environmental Research and Public Health. Generally, the topic fits into the scope of the journal and it respects Scientific Best Practice in terms of the structure. However, the manuscript has several issues that require revision. Beside this, it is nessary to carefully cross check the content and figures regarding similarities in the paper "Exploring Natural Models for the 'Rolling Unmasking Effect' of Downwind Odor Dispersion; Prairie Verbena, Prehensile-Tailed Porcupine and Virginia Pepperweed", published in 2018, were even are equal figures are contained.

The introduction must be revised throughout as it is written like a research report and doesnt represent the style of a scientific article. Parts of the section called "background" can be used to provide content for the introduction, and some of the content that is currently in the introduction section should be moved to the end of the literature review section as conclusion of the literature review and the identified scientific gaps that require the present investigation.

Moreover, I recommend to include a flow charts into the section materials and methods in order to illustrate the methodology graphically,

The reasons for the selection of the investigation location are unclear, as well as the uncertainties of the investigation. Why was selected the porcupine?

From my point of view, is not necessary to add supplementary material, but to tighten up both documents, leave out unrelevant information, and to merge them.

Regarding the results and assessment, for the reviewer remains unclear how odours of other sources are devided, for instance for the phenols like cresol. There are several other sources that can emit phenols.

The sense of figure 9 is completely unclear. It appears non-scientific.

The conclusions are very general and must be improved. In the conclusions, in addition to summarising the actions taken and results, please strengthen the explanation of their significance. It is recommended to use quantitative reasoning comparing with appropriate benchmarks, especially those stemming from previous work.

Reviewer 2 Report

This paper by Wright et al. describes a very innovative rolling unmasking effect approach to illustrate and deconvolute naturally occurring odorous phenomena from the environment. The research work is innovative, well-described, and organized. The potentials of this work are high especially in the field of monitoring and management of environmental odors. I suggest the publication of the manuscript in the current state. 

supplementary:

Strengths of the paper: 1. The originality and novelty of the study itself 2. The in-depth experimental investigation done by the authors 3. The wide applicability of the proposed approach and model in real-life

Weakness of the paper: 1. In the introduction section, the authors could expand their literature survey and could discuss the use of portable and miniature mass spectrometry in odor detection, monitoring, and localization, e.g. work done by Prof. Guido Verbeck, Prof. Stephen Taylor, Prof. Cooks. On-line chemical analysis with real-time monitoring capabilities using mobile mass spectrometry has been previously used to locate illicit production of drugs of abuse (Verbeck et al.) 2. The paper and the readers could also benefit if some statistical analysis of the results would be presented, e.g. chemometrics analysis, e.g. principal component analysis, ASCA, etc. of the odorant molecules detected.  

Reviewer 3 Report

The authors proposed downwind environmental odor prioritization approach and explored rolling unmasking effect. The proposed approach utilized a panel testing and analytical instrument testing, and integrate them to identify/prioritize the animal odor. The authors well presented the procedure and results, but it is slightly misaligned with this journal’s topic. Thus, the authors should revise the manuscript to cover the interests of this journal and its audiences. Please see below comments;

Introduction: The relationship between odor problem and public health is weak. Given this journal’s topic, please add a paragraph stating the public health harm of odor problem, and how this approach can handle it.

Line 181: Please refer what is GC-O. All abbreviations should be spelled out and explained at first use.

Line 210: This section can move to the end of method section. So, the authors don’t need to repeat technical details such as SPME fiber type, etc.

Line 218: Please specify injection duration (or desorption duration).

Table 2: The authors may pick up some of these chemicals which are irritants and/or allergens, and discuss how to mitigate odor and health risk problems using the proposed approach.

Conclusions: Please add one or two sentences of public health implication message(s) concluded from above comments.

Author Response

Response to Reviewer 3

Comments and Suggestions for Authors

The authors proposed downwind environmental odor prioritization approach and explored rolling unmasking effect. The proposed approach utilized a panel testing and analytical instrument testing, and integrate them to identify/prioritize the animal odor. The authors well presented the procedure and results, but it is slightly misaligned with this journal’s topic. Thus, the authors should revise the manuscript to cover the interests of this journal and its audiences. Please see below comments;

Introduction: The relationship between odor problem and public health is weak. Given this journal’s topic, please add a paragraph stating the public health harm of odor problem, and how this approach can handle it.

Author Response: We did not specifically add new narrative in the Introduction on the possible  relationship between P.T. porcupine odor and health. This is outside the bounds of the process demonstration scope of the project. However, the following statement has been added to the Conclusions to emphasize that the proposed process could provide value to the community discussion regarding odor and environmental health issues, should they exist.

‘Since the P.T. porcupine was selected as a ‘neutral’ demonstration of the odorant prioritization process, it does not reflect an actual community environmental odor issue.  However, if it had been, it is noteworthy that the downwind citizenry could have been: (1) made aware that the proposed impact-priority ‘onion’ odorants were present at levels which are below the detection limits of one of our most-sensitive electronic detectors; (2) these impact-priority odorants are also common to onion emission and therefore unlikely to have a high toxicological impact and (3) given the opportunity to confirm for themselves, through odor-matching demonstration utilizing their own sensory capabilities, that the proposed impact-priority hypothesis are correct. This possibility has the potential for alleviating some psychologically induced health effects, should they exist.’

Line 181: Please refer what is GC-O. All abbreviations should be spelled out and explained at first use.

Author Response: we fixed it.

Line 210: This section can move to the end of method section. So, the authors don’t need to repeat technical details such as SPME fiber type, etc.

 Author Response: we agree with the Reviewer. We rearranged the order of the entire Materials and Methods section to the ‘sampling’-‘separation’ – ‘analysis/chemical identification’ order.

Line 218: Please specify injection duration (or desorption duration).

Author Response: A characteristic of SPME fiber injection is that the injection duration is the same as the analysis duration.  The SPME fibers were left in place once they were inserted; achieving: (i) the initial adsorbed VOC injection for the current analysis and (ii) preconditioning the fiber for the subsequent sample collection event. We added this information to the manuscript in Methods.

Table 2: The authors may pick up some of these chemicals which are irritants and/or allergens, and discuss how to mitigate odor and health risk problems using the proposed approach.

Author Response: We feel that this effort, through odorant prioritization, is focused specifically on strategies for defining the specific chemicals which are primarily responsible for a targeted environmental odor.  Therefore the more general questions regarding health impact of a larger field of chemicals, whether odorous or non-odorous, is outside the scope of this effort. Since the the P.T. porcupine was selected as a ‘neutral’ demonstration of the odorant prioritization process, it does not reflect an actual community environmental odor issue.  However, if it had been, it is noteworthy that the downwind citizenry could have been: (1) made aware that the proposed impact-priority ‘onion’ odorants were present at levels which are below the detection limits of one of our most-sensitive electronic detectors; (2) these impact-priority odorants are also common to onion emission and therefore unlikely to have a high toxicological impact and (3) given the opportunity to confirm for themselves, through odor-matching demonstration utilizing their own sensory capabilities, that the proposed impact-priority hypothesis are correct. This possibility has the potential for alleviating some psychologically induced health effects; should they exist.  We added this in the Conclusion (also in response to Reviewer 1), see below.   

Conclusions: Please add one or two sentences of public health implication message(s) concluded from above comments.

Author Response: The following statement has been added to the Conclusion to emphasize the connection between odor and environmental health.

‘Since the P.T. porcupine was selected as a ‘neutral’ demonstration of the odorant prioritization process, it does not reflect an actual community environmental odor issue.  However, if it had been, it is noteworthy that the downwind citizenry could have been: (1) made aware that the proposed impact-priority ‘onion’ odorants were present at levels which are below the detection limits of one of our most-sensitive electronic detectors; (2) these impact-priority odorants are also common to onion emission and therefore unlikely to have a high toxicological impact and (3) given the opportunity to confirm for themselves, through odor-matching demonstration utilizing their own sensory capabilities, that the proposed impact-priority hypothesis is correct. This possibility has the potential for alleviating some psychologically induced health effects, should they exist.’    

Round 2

Reviewer 1 Report

Thank you for providing the revised version. More or less all my comments have been addressed. From my point of view, the literature review is still weak and should be supported with more scientific references.

Author Response

Response to Reviewer 1

Comments and Suggestions for Authors

Thank you for providing the revised version. More or less all my comments have been addressed. From my point of view, the literature review is still weak and should be supported with more scientific references.

Author’s Response:

We are thankful for the opportunity to improve the literature review and to strengthen the manuscript. The following additions were made:

  1. We added four additional references [6,7,14,15]:
    1. Bokowa, A.; Diaz, C.; Koziel, J.A.; McGinley, M.; Barclay, J.; Schauberger, G.; Guillot, J.M.; Sneath, R.; Capelli, L.; Zorich, V.; Izquierdo, C.; Bilsen, I.; Romain, A.C.; Cabeza, M.; Liu, D.; Both, R.; Belois, H.; Higuchi, T.; Wahe, L. Summary and overview of odour regulations worldwide: Atmosphere 2021, 12, 206. doi.org/10.3390/atmos12020206. [new reference 6]
    2. Brancher, M.; Griffiths, K.D.; Franco, D.; de Melo Lisboa, H. A review of odour impact criteria in selected countries around the world. Chemosphere 2017, 168, 1531-1570. doi.org/10.1016/j.chemosphere.2016.11.160. [new reference 5]
    3. Koziel, J.A.; Guenther, A.; Byers, M.; Iwasinska, A.; Parker, D.; Wright, D. Update on the development of a new ASTM standard for environmental odor assessment. Plenary Lecture at the NOSE2020, 7th International Conference on Environmental Odour Monitoring and Control. Virtual Conference; 18-21 April 2021. [new reference 14]
    4. E. Byers Scientific, Iowa State University, and odor experts identify the volatile chemical compound responsible for cannabis odor complaints. Press Release on March 22, 2021. Business Wire (a Berkshire Hathaway Company), Bloomington, Indiana, USA. (Available online: https://www.businesswire.com/news/home/20210322005837/en/Byers-Scientific-Iowa-State-University-and-Odor-Experts-Identify-the-Volatile-Chemical-Compound-Responsible-for-Cannabis-Odor-Complaints; accessed on 29 November 2021). [new reference 15]
  2. The following new edits were made:
    1. Section 2.1 – “In reviews of international odor regulations [6,7] it was reported that almost all standards are based upon odor concentration limits by forced-choice olfactometry, reflecting either; (a) laboratory olfactometer based odour concentration units per cubic meter; (b) triangle bag-based odour index threshold value measurement or (c) [6] field olfactometry-based offensiveness measurement. There were no major international entities which referenced the use of chemical-analysis based methods (i.e. GC-MS, GC-Olfactometry, odorant prioritization) for environmental odor assessment, monitoring or mitigation.”
    2. Section 2.1 – “In a more recent study [14,15], these authors were able to identify the specific chemical odorant which is believed primarily responsible for the reported ‘skunky’ odor downwind of dense cannabis-growing operations. The team utilized an analytical approach (air sampling with solid-phase microextraction, SPME; and analysis on a gas chromatography – mass spectrometry – olfactometry system, GC-MS-O), leaf enclosure study and field observation, to isolate, identify, measure and ultimately conclude that the compound 3-methyl-2-butene-1-thiol (i.e., 321 MBT), was the primary source of this ‘skunky’ odor of cannabis [14,15]. Historically, this ‘skunky’ downwind odor has often been tied to terpenes. The 321 MBT reported discovery as the actual link with ‘skunky’ cannabis sup-ports the more persuasive expectation of a sulfur component within the emission profile of cannabis [14,15].”
    3. Section 1.2 – “The novelty proposed herein lies in identification of those few compounds responsible for the downwind odor impacts and requiring mitigation focus.”

Reviewer 3 Report

The authors well addressed the comments. I recommend accept without further comments.

Author Response

We are thankful for the comments. We still made additional improvements to the literature review as requested by another Reviewer The following additions were made:

  1. We added four additional references [6,7,14,15]:
    • Bokowa, A.; Diaz, C.; Koziel, J.A.; McGinley, M.; Barclay, J.; Schauberger, G.; Guillot, J.M.; Sneath, R.; Capelli, L.; Zorich, V.; Izquierdo, C.; Bilsen, I.; Romain, A.C.; Cabeza, M.; Liu, D.; Both, R.; Belois, H.; Higuchi, T.; Wahe, L. Summary and overview of odour regulations worldwide: Atmosphere 2021, 12, 206. doi.org/10.3390/atmos12020206. [new reference 6]
    • Brancher, M.; Griffiths, K.D.; Franco, D.; de Melo Lisboa, H. A review of odour impact criteria in selected countries around the world. Chemosphere 2017, 168, 1531-1570. doi.org/10.1016/j.chemosphere.2016.11.160. [new reference 5]
    • Koziel, J.A.; Guenther, A.; Byers, M.; Iwasinska, A.; Parker, D.; Wright, D. Update on the development of a new ASTM standard for environmental odor assessment. Plenary Lecture at the NOSE2020, 7th International Conference on Environmental Odour Monitoring and Control. Virtual Conference; 18-21 April 2021. [new reference 14]
    • E. Byers Scientific, Iowa State University, and odor experts identify the volatile chemical compound responsible for cannabis odor complaints. Press Release on March 22, 2021. Business Wire (a Berkshire Hathaway Company), Bloomington, Indiana, USA. (Available online: https://www.businesswire.com/news/home/20210322005837/en/Byers-Scientific-Iowa-State-University-and-Odor-Experts-Identify-the-Volatile-Chemical-Compound-Responsible-for-Cannabis-Odor-Complaints; accessed on 29 November 2021). [new reference 15]
  2. The following new edits were made:
    • Section 2.1 – “In reviews of international odor regulations [6,7] it was reported that almost all standards are based upon odor concentration limits by forced-choice olfactometry, reflecting either; (a) laboratory olfactometer based odour concentration units per cubic meter; (b) triangle bag-based odour index threshold value measurement or (c) [6] field olfactometry-based offensiveness measurement. There were no major international entities which referenced the use of chemical-analysis based methods (i.e. GC-MS, GC-Olfactometry, odorant prioritization) for environmental odor assessment, monitoring or mitigation.”
    • Section 2.1 – “In a more recent study [14,15], these authors were able to identify the specific chemical odorant which is believed primarily responsible for the reported ‘skunky’ odor downwind of dense cannabis-growing operations. The team utilized an analytical approach (air sampling with solid-phase microextraction, SPME; and analysis on a gas chromatography – mass spectrometry – olfactometry system, GC-MS-O), leaf enclosure study and field observation, to isolate, identify, measure and ultimately conclude that the compound 3-methyl-2-butene-1-thiol (i.e., 321 MBT), was the primary source of this ‘skunky’ odor of cannabis [14,15]. Historically, this ‘skunky’ downwind odor has often been tied to terpenes. The 321 MBT reported discovery as the actual link with ‘skunky’ cannabis sup-ports the more persuasive expectation of a sulfur component within the emission profile of cannabis [14,15].”
    • Section 1.2 – “The novelty proposed herein lies in identification of those few compounds responsible for the downwind odor impacts and requiring mitigation focus.”

This manuscript is a resubmission of an earlier submission. The following is a list of the peer review reports and author responses from that submission.

Round 1

Reviewer 1 Report

The article explores different problems such as odour characterisation, odour prioritisation and rolling unmasking effect (RUE) using three case studies: two vegetal sources and one animal source. The direct sources of odour were sampled and different sampling conditions were taken into account. Appropriate analytical methods were used for profile identification. “Simulated” dilution through sampling treatment was used to “recreate” dilution due to downwind dispersion from the odour source. This “simulated” dilutions may represent a qualitative demonstration of RUE, however there were no references to compare with, so sound conclusions cannot be made on this regard.

Better detail on the definition of the odor characters and the subjects participating in the odour sniff tests is missing.

No repeated measurements were specified nor reproducibility of the tests performed was mentioned. I suggest the authors include comparative tables containing the conditions of experiments, samplings and results to improve communication of results, beyond the qualitative aspect of it (e.g, including peak areas and uncertainties to understand which are actually representative)

Quality of chromatograms and other diagrams can be improved/standardised.

Author Response

Response to Reviewer 1

Yes

Can be improved

Must be improved

Not applicable

Does the introduction provide sufficient background and include all relevant references?

( )

(x)

( )

( )

Is the research design appropriate?

( )

( )

(x)

( )

Are the methods adequately described?

(x)

( )

( )

( )

Are the results clearly presented?

( )

(x)

( )

( )

Are the conclusions supported by the results?

( )

(x)

( )

( )

Comments and Suggestions for Authors

Reviewer: The article explores different problems such as odour characterisation, odour prioritisation and rolling unmasking effect (RUE) using three case studies: two vegetal sources and one animal source. The direct sources of odour were sampled and different sampling conditions were taken into account. Appropriate analytical methods were used for profile identification. “Simulated” dilution through sampling treatment was used to “recreate” dilution due to downwind dispersion from the odour source.

Author’s Response: we are thankful for constructive feedback on the manuscript. We addressed all comments and strengthened the manuscript. Changes are tracked. The statement above is a very succinct summary of the manuscript.

Reviewer: This “simulated” dilutions may represent a qualitative demonstration of RUE, however there were no references to compare with, so sound conclusions cannot be made on this regard.

Author’s Response:

  • We agree with the Reviewer. This concept is qualitative by nature. As a first step, we are looking for a ‘low-hanging fruit’, ‘single odor-defining compounds’. Please see below for the added definitions and description of the whole proposed process. In this case (i.e., first step), no quantitative or statistical data should be required at this stage of the investigative process (description added in section 3.1).
  • We revised the manuscript accordingly by:
    • Adding ‘Qualitative’ to the title.
    • We added the new section in the Methods (3.1. Odorant Prioritization Procedural Summary Outline) that describes the 7 steps of which the first 5 (illustrated in this manuscript) are qualitative. We make sure the description of steps contains the ‘qualitative’ keyword. Steps 6 and 7 are also defined here as quantitative and subsequent after this qualitative first step assessment.
    • Many odor-solving attempts fail because they start with a quantitative (e.g., Steps 6 & 7) approach. Starting with the analytical approach can be frustratingly futile, e.g., generating a massive amount of data, which does not have much value/impact relative to solving odor issues.
    • In the case of P.T. porcupine described in the manuscript - we did complete an informal execution of Steps 4-5, which is odor-matching-based validation with additional panelists.
    • The Results section has a new Table 1 that serves as a reference guide of which steps are completed and illustrated for each case.
  • We added a narrative to Conclusions which emphasizes the advantage that the 3 selected natural models carry:
    • ‘(a) each is publicly accessible to the readership audience (with varying degrees of accessibility); (b) most of the readership audience is equipped with sensory abilities that are equal to or better than those of the lead investigator and collaborators and (c) therefore can perform their own assessment and determine if they are in agreement with the odor-character descriptions of the collaborators.

Reviewer: Better detail on the definition of the odor characters and the subjects participating in the odour sniff tests is missing.

Author’s Response: We addressed this by:

  • The definition of the ‘odor characters’, i.e., ‘what it smells like’ is already in the manuscript (Section 1.2, line 66 in ‘simple markup’ mode).
  • Per ‘subjects participating’ – we added Tables 2, 3, 4 that identify participating subjects (as co-authors or associates) and their responses to questions about odor descriptor and odor-match (yes/no). However, it is intrinsic to this proposed approach that the principal investigator makes the initial assessment and takes the lead of odorant prioritization in Steps 1-3. This person needs to be highly experienced with odor troubleshooting. We added this information to the definition of Steps (Section 3.1):
    • ‘Step 1 – Downwind composite odor assessment –qualitative, at-site odor-character assessment by the panelist; in this case, (D.W.), a GC-O investigator with 20+ years of experience odor troubleshooting for industry. The goal of this stage is to observe recognizable odors that are consistent and perceived as characteristic of the source, at the downwind outer boundary, and at the time of the at-site assessment.
    • Step 2 - GC-O-based odorant prioritization – qualitative, on-instrument assessment by the panelist, (D.W.); attempting to make a connection between the observed downwind odor character and individual compounds which are perceived as character-defining for that odor.
    • Step 3 – First-pass odor-match validation of impact-priority hypothesis from Step 2- qualitative odor-match based confirmation by conference with associate GC-O investigator(s), where possible; in the case of the Virginia pepperweed, an experienced (A.I.) GC-O investigator with 10+ years of experience odor troubleshooting for industry; generally involving on-instrument GC-O based crosscheck.
    • Step 4 – Development of a synthetic formulation for final odor-match-based validation - the panelist, (D.W.) attempts to develop a formulation, in low odor, food-grade propylene glycol carrier which reflects a high-fidelity odor-match to that of the targeted environment downwind. This formulation can range from very simple single odorants to multi-odorant blends, matching the odorant concentration ratios existing in the targeted environments downwind.
    • Step 5 – Final odor-match validation of impact-priority hypothesis from Steps 1 and 2 - qualitative or quantitative odor-match-based validation by conference with volunteer sensory panelists drawn from (a) downwind citizenry; (b) other community stakeholders or (c) professional sensory panel.
    • Step 6 – Analytical method development targeting impact-priority odorants defined and validated; Steps 1 through 5 – quantitative method development for follow-on odor investigation, monitoring, and mitigation strategy focusing.
    • Step 7 – Instrument-based environmental odor monitoring based upon impact-priority odorants – quantitative monitoring for correlating downwind environmental impact and upwind source prioritization.’

Reviewer: No repeated measurements were specified nor reproducibility of the tests performed was mentioned. I suggest the authors include comparative tables containing the conditions of experiments, samplings and results to improve communication of results, beyond the qualitative aspect of it (e.g, including peak areas and uncertainties to understand which are actually representative)

Author’s Response: We addressed this by:

  • we added Tables 2, 3, 4 in the Results that identify participating subjects (as co-authors or associates) and their responses to questions about odor descriptor, odor-match (yes/no), match-source, and odor-match-components.
  • We emphasized the qualitative approach that must occur before quantitative assessment as described in the newly-added definitions and descriptions of Steps 1-7.

Reviewer: Quality of chromatograms and other diagrams can be improved/standardised.

Author’s Response: We replaced all images and chromatograms with higher-quality versions. We also added photos of prairie verbena, P.T. porcupine, and Virginia pepperweed as Figures S1-S5 in Supplementary Material.

Reviewer 2 Report

Introduction. The introduction is very long and contains a lot of explanatory information. It looks a bit like a textbook rather than an introduction to a scientific paper.

Figure 1 is not clear.

Result: Perhaps, for a better understanding of the experiment, it is worthwhile to first give its diagram?

In conclusion, it is worth describing how the work will develop further.

References: it is not recommended to use multiple links, for example [16-19]. It is recommended to relate the link to specific information. And if there is something important in the cited paper, it is recommended to mark it separately.

Author Response

Response to Reviewer 2

Yes

Can be improved

Must be improved

Not applicable

Does the introduction provide sufficient background and include all relevant references?

( )

(x)

( )

( )

Is the research design appropriate?

( )

(x)

( )

( )

Are the methods adequately described?

( )

(x)

( )

( )

Are the results clearly presented?

( )

(x)

( )

( )

Are the conclusions supported by the results?

( )

(x)

( )

( )

Comments and Suggestions for Authors

Reviewer: Introduction. The introduction is very long and contains a lot of explanatory information. It looks a bit like a textbook rather than an introduction to a scientific paper.

Author's Response: we are thankful for constructive feedback on the manuscript. We addressed all comments and strengthened the manuscript. Changes are tracked.

We addressed this by revising the content, creating separate (and shorter) Introduction and Background sections. Therefore, readers would be able to skip the Background if not needed.

Reviewer: Figure 1 is not clear.

Author's Response: we revised the caption for Figure 1 to:

'Pictorial representation of the 'rolling unmasking effect' (RUE). The source is a complex mixture of odorants (left), yet it is simplified to a single impactful odorant (illustrated with the yellow dot) at the receptor (right) downwind. The odor frontal boundary represents the farthest downwind reach (impact; marked with a yellow oval) of a single compound (marked with yellow circle at the source and receptors' nose), while the internal colored ovals represent the boundaries of sequential odor unmasking as the secondary-impact odorants are diluted below their detection/masking concentration levels.'

In addition, we have improved the quality of all figures in the manuscript, and added photos of prairie verbena, P.T. porcupine, and Virginia pepperweed as Figures S1-S5 in Supplementary Material.

Reviewer: Result: Perhaps, for a better understanding of the experiment, it is worthwhile to first give its diagram?

Author's Response:

  • We added Table 1 that serves as a diagram/summary of the cases described in the Results. Table 1 contains the Steps 1-7 short description and list of Figures illustrating steps for each case 1-3. We believe it will be helpful to readers and serve as a quick guide to Results.
  • We added the new section in the Methods (3.1. Odorant Prioritization Procedural Summary Outline) that describes the 7 steps of which the first 5 (illustrated in this manuscript) are qualitative. We make sure the description of steps contains the 'qualitative' keyword. Steps 6 and 7 are also defined here as quantitative and subsequent after this qualitative first step assessment.
  • We added this information to the definition of Steps:
    • 'Step 1 – Downwind composite odor assessment –qualitative, at-site odor-character assessment by the panelist; in this case, (D.W.), a GC-O investigator with 20+ years of experience odor troubleshooting for industry. The goal of this stage is to observe recognizable odors that are consistent and perceived as characteristic of the source, at the downwind outer boundary and at the time of the at-site assessment.
    • Step 2 - GC-O-based odorant prioritization – qualitative, on-instrument assessment by the panelist, (D.W.); attempting to make a connection between the observed downwind odor character and individual compounds which are perceived as character-defining for that odor.
    • Step 3 – First-pass odor-match validation of impact-priority hypothesis from Step 2- qualitative odor-match based confirmation by conference with associate GC-O investigator(s), where possible; in the case of the Virginia pepperweed, an experienced (A.I.) GC-O investigator with 10+ years of experience odor troubleshooting for industry; generally involving on-instrument GC-O based crosscheck.
    • Step 4 – Development of a synthetic formulation for final odor-match-based validation - the panelist, D.W.) attempts to develop a formulation, in low odor, food-grade propylene glycol carrier which reflects a high-fidelity odor-match to that of the targeted environment downwind. This formulation can range from very simple single odorants to multi-odorant blends, matching the odorant concentration ratios existing in the targeted environments downwind.
    • Step 5 – Final odor-match validation of impact-priority hypothesis from Steps 1 and 2 - qualitative or quantitative odor-match-based validation by conference with volunteer sensory panelists drawn from (a) downwind citizenry; (b) other community stakeholders or (c) professional sensory panel.
    • Step 6 – Analytical method development targeting impact-priority odorants defined and validated; Steps 1 through 5 – quantitative method development for follow-on odor investigation, monitoring, and mitigation strategy focusing.
    • Step 7 – Instrument-based environmental odor monitoring based upon impact-priority odorants – quantitative monitoring for correlating downwind environmental impact and upwind source prioritization.'

Reviewer: In conclusion, it is worth describing how the work will develop further.

Author's Response: We described the whole process, next steps in the manuscript.   

  • We added the new section in the Methods (3.1. Odorant Prioritization Procedural Summary Outline) that describes the 7 steps of the whole process.
  • We are finalizing the next manuscript (quantitative approach). The early version is posted in Preprints:
    • Wright, D.; Koziel, J.; Parker, D.; Iwasinska, A. Part 2: Odor-Cued Grab Sampling of Transient Environmental Odor Events; Mapping the 'Rolling Unmasking Effect' of Downwind Odor Dispersion. Preprints 2020, 2020080520 (doi: 20944/preprints202008.0520.v1).
  • This manuscript (qualitative) and the next (quantitative) are proposed as prominent demonstration references for our ongoing efforts within ASTM to develop a new Standard Practice for environmental odor assessment utilizing SPME/MDGC-MS-O based odorant prioritization. Several co-authors (D.W., J.K. D.P) work within the ASTM WK72782 workgroup to develop this proposed Standard Practice. It is work-in-progress, not a standard yet.

Reviewer: References: it is not recommended to use multiple links, for example [16-19]. It is recommended to relate the link to specific information. And if there is something important in the cited paper, it is recommended to mark it separately.

Author's Response: we revised the manuscript and parsed the narrative to provide the context and content of separate references.

Round 2

Reviewer 1 Report

I have been reviewing the article and was trying to go over the changes authors did to the manuscript (which is now more than 40pp).

It have been difficult to me to give better feedback than the one I did for the first review, since I still do not think discussion or conclusions are sound enough to improve the current state of the art.